# Entropy Penalty: Towards Generalization Beyond the IID Assumption

## Abstract

It has been shown that instead of learning actual object features, deep networks tend to exploit non-robust (spurious) discriminative features that are shared between training and test sets. Therefore, while they achieve state of the art performance on such test sets, they achieve poor generalization on out of distribution (OOD) samples where the IID (independent, identical distribution) assumption breaks and the distribution of non-robust features shifts. Through theoretical and empirical analysis, we show that this happens because maximum likelihood training (without appropriate regularization) leads the model to depend on all the correlations (including spurious ones) present between inputs and targets in the dataset. We then show evidence that the information bottleneck (IB) principle can address this problem. To do so, we propose a regularization approach based on IB called Entropy Penalty, that reduces the model's dependence on spurious features–features corresponding to such spurious correlations. This allows deep networks trained with Entropy Penalty to generalize well even under distribution shift of spurious features. As a controlled test-bed for evaluating our claim, we train deep networks with Entropy Penalty on a colored MNIST (C-MNIST) dataset and show that it is able to generalize well on vanilla MNIST, MNIST-M and SVHN datasets in addition to an OOD version of C-MNIST itself. The baseline regularization methods we compare against fail to generalize on this test-bed.

## 1 Introduction

It is now known that deep networks trained on clean training data (without proper regularization) often learn spurious (non-robust) features which are features that can discriminate between classes but do not align with human perception (Jo & Bengio, 2017; Geirhos et al., 2018a; Tsipras et al., 2018; Ilyas et al., 2019). An example of non-robust feature is the presence of desert in camel images, which may correlate well with this object class. More realistically, models can learn to exploit the abundance of input-target correlations present in datasets, not all of which may be invariant under different environments. Interestingly, such classifiers can achieve good performance on test sets which share the same non-robust features. However, due to this exploitation, these classifiers perform poorly under distribution shift (Geirhos et al., 2018a; Hendrycks & Dietterich, 2019) because it violates the IID assumption which is the foundation of existing generalization theory (Bartlett & Mendelson, 2002; McAllester, 1999b;a).

The research community has approached this problem from different directions. In part of domain adaptation literature (Eg. Ganin & Lempitsky (2014)), the goal is to adapt a model trained on a source domain (often using unlabeled data) so that its performance improves on a target domain that contains the same set of target classes but under a distribution shift. There has also been research on causal discovery (Hoyer et al., 2009; Janzing et al., 2009; Lopez-Paz et al., 2017; Kilbertus et al., 2018) where the problem is formulated as identifying the causal relation between random variables. This framework may potentially then be used to train a model that only depends on the relevant features. However, it is often hard to discover causal structure in realistic settings. Adversarial training (Goodfellow et al., 2014; Madry et al., 2017) on the other hand aims to learn models whose predictions are invariant under small perturbations that are humanly imperceptible. Thus adversarial training can be seen as the worst-case distribution shift in the local proximity of the original training distribution.

Our goal is different from the aforementioned approaches. We aim to directly learn a classification model using labeled data which is capable of generalizing well under input distribution shift (not constrained to being locally) without making any changes to the model during test time. Thus our goal is more aligned with the recently proposed Invariant Risk Minimization (Arjovsky et al., 2019), but imposes less constraints on the data collection process. For a detailed discussion on related work, see section 4. Our contributions are as follows:

1. Our theoretical and empirical analysis shows that models trained with maximum likelihood objective without appropriate regularization can, in general, learn to exploit/depend on all the correlations present between inputs and targets in the training set, leading to non-robust representations. While this representation may allow them to yield state-of-the-art performance on test sets whose distribution is identical to the training set, they would perform poorly when the distribution of non-robust features shift. This effect is not mitigated by a larger training set containing more variations between samples. Thus based on our analysis, it should not be surprising that deep networks trained on image datasets show poor performance under input perturbations and (in general) input distribution shifts as discussed in numerous recent papers (Hendrycks & Dietterich, 2019; Jo & Bengio, 2017; Geirhos et al., 2018b;a).

2. We provide evidence showing that the information bottleneck (IB) principle (Tishby et al., 2000; Tishby & Zaslavsky, 2015) is capable of addressing the out of distribution generalization problem. Specifically, our proposal, that we call *Entropy Penalty* is based on the IB principle and aims at learning a representation that throws away maximum possible information about the input distribution while achieving the goal of correctly predicting targets. Intuitively, doing so makes the representation agnostic to non-robust features in the input, thus allowing the model's predictions to be invariant under a shift of the distribution of such features during test time.

3. We show experimental results using Entropy Penalty in which a deep network trained on a colored version of MNIST dataset (see appendix A for samples) is able to generalize well on vanilla MNIST, MNIST-M, SVHN and a distribution-shifted version of the colored MNIST dataset itself. We note that most of the baseline methods failed to the extent of achieving performance close to random chance.

## 2 How to Generalize under Distribution Shift?

In order to approach a solution to our problem, we observe that the out of distribution nature of samples during test time arises because certain aspects of the input change even though the input still corresponds to the same pool of targets as seen during training. An instance of this change would be seeing a camel in the city (during test time) which has dramatically different background features compared to the desert (seen during training). Thus, if a model is trained to depend only on camel features (which defines the class more universally) and ignore other aspects, a shift in the distribution of such aspects will no longer affect the model's decision during test time.

We note that the above intuition is encapsulated by the information bottleneck learning principle (Tishby et al., 2000; Tishby & Zaslavsky, 2015) which minimizes the following objective,

$$\mathcal{L}_{IB}(\theta) = -\mathcal{I}(f_\theta(\mathbf{X}), \mathbf{Y}) + \beta \mathcal{I}(f_\theta(\mathbf{X}), \mathbf{X}) \tag{1}$$

where $\mathbf{X}$ and $\mathbf{Y}$ represent the input and target (often class label) random variables, $f_\theta(.)$ denotes a deterministic model with learnable parameters $\theta$, and $\beta$ is a hyper-parameter. While the first term effectively maximizes the training data likelihood, it is the second term that regularizes the model representations to become invariant to non-robust features that are not dominantly present in all samples by minimizing mutual information between input and representation random variables. We now derive Entropy Penalty which is equivalent to the IB regularization for deterministic models. We note that $\mathcal{I}(f_\theta(\mathbf{X}), \mathbf{Y})$ and $\mathcal{I}(f_\theta(\mathbf{X}), \mathbf{X})$ can be written equivalently as follows,

$$\mathcal{I}(f_\theta(\mathbf{X}), \mathbf{Y}) = \mathcal{H}(\mathbf{Y}) - \mathcal{H}(\mathbf{Y}|f_\theta(\mathbf{X})) \tag{2}$$
$$\mathcal{I}(f_\theta(\mathbf{X}), \mathbf{X}) = \mathcal{H}(f_\theta(\mathbf{X})) - \mathcal{H}(f_\theta(\mathbf{X})|\mathbf{X}) \tag{3}$$

where $\mathcal{H}(.)$ denotes entropy and $\mathcal{H}(.|.)$ denotes conditional entropy. Note that $\mathcal{H}(\mathbf{Y})$ in Eq. 2 is fixed and $\mathcal{H}(\mathbf{Y}|f_\theta(\mathbf{X}))$ is the same as the maximum likelihood loss. Secondly, since $f_\theta(.)$ is a deterministic function, the conditional entropy $\mathcal{H}(f_\theta(\mathbf{X})|\mathbf{X})$ is fixed. Thus we only need to minimize

the entropy of the learned representation $\mathcal{H}(f_\theta(\mathbf{X}))$ in Eq. 3. Thus in the case of deterministic $f_\theta(.)$, the minimization in IB objective can be equivalently framed as,

$$\hat{\mathcal{L}}_{IB}(\theta) = \mathcal{H}(\mathbf{Y}|f_\theta(\mathbf{X})) + \beta\mathcal{H}(f_\theta(\mathbf{X})) \tag{4}$$

Therefore, we call this general form of regularization *Entropy Penalty*. Note that while the first term (data likelihood) is easy to optimize, the second term is often intractable for continuous high dimensional distributions. The following proposition shows an equivalent form of the above objective.

**Proposition 1** $\hat{\mathcal{L}}_{IB}(\theta) = (1-\beta)\mathcal{H}(\mathbf{Y}|f_\theta(\mathbf{X})) + \beta\mathcal{H}(f_\theta(\mathbf{X})|\mathbf{Y}) + C$

*where $C$ is a positive constant for discrete $\mathbf{Y}$, independent of $\theta$.*

The benefit of the above form for $\hat{\mathcal{L}}_{IB}(\theta)$ instead of Eq. 4 is that it can often be easier to model the conditional $\Pr(f_\theta(\mathbf{X})|\mathbf{Y})$ compared to the marginal $\Pr(f_\theta(\mathbf{X}))$. For instance, if we assume $\Pr(f_\theta(\mathbf{X})|\mathbf{Y})$ is Gaussian for all class labels $\mathbf{Y}$, the entropy of the conditional distribution $\Pr(f_\theta(\mathbf{X})|\mathbf{Y})$ has a closed-form solution given by $0.5\log(2\pi e\sigma_\mathbf{Y}^2)$, where $\sigma_\mathbf{Y}^2$ denotes the variance of the class conditional Gaussian distribution of $f_\theta(\mathbf{X})$ for class $\mathbf{Y}$.

On a practical note, when applying entropy penalty to deep networks, we found that it was not effective when applied to the last layer representations. However, applying this penalty to the first hidden layer representation improved performance under distribution shift. While we do not have a complete explanation for this behavior, we conjecture that this could be because of the data processing inequality for deep networks (Tishby & Zaslavsky, 2015) which states,

$$\mathcal{I}(\mathbf{h}_1, \mathbf{X}) \geq \mathcal{I}(\mathbf{h}_2, \mathbf{X}) \geq \ldots \geq \mathcal{I}(\mathbf{h}_L, \mathbf{X}) \tag{5}$$

where $\mathbf{h}_i$ denotes the $i^{th}$ hidden layer representation and $L$ is the depth of the network. Writing mutual information in terms of entropy and conditional entropy, and taking advantage of the fact that the conditional entropy term is fixed for a deterministic conditional, we have that,

$$\mathcal{H}(\mathbf{h}_1) \geq \mathcal{H}(\mathbf{h}_2) \geq \ldots \geq \mathcal{H}(\mathbf{h}_L) \tag{6}$$

Thus entropy is larger for lower layers. Further, minimizing entropy for higher layers does not ensure entropy is minimized for lower layers due to the above inequalities. Thus, any excess information about the input captured by the first layer gets propagated to the higher layers, the effect of which may get amplified under distribution shift if entropy minimization at last layer is not done appropriately. For all experiments conducted using entropy penalty (EP), we use the aforementioned Gaussian assumption on the representation of the first hidden layer and compute its class conditional variance in order to minimize the conditional entropy. Specifically, let $\mathbf{h}(\mathbf{x})$ represent the first hidden layer representation of an input $\mathbf{x}$ (before non-linearity), then we implement EP as,

$$\mathcal{R}_{EP} = \sum_{k=1}^K \mathbb{E}_{\mathbf{x}\sim\mathcal{D}(\mathbf{x}|y=k)}[(\mathbf{h}(\mathbf{x}) - \mu_k)^2] \tag{7}$$

where $\mu_k := \mathbb{E}_{\mathbf{x}\sim\mathcal{D}(\mathbf{x}|y=k)}[(\mathbf{h}(\mathbf{x})]$ and $\mathcal{D}$ denotes the data distribution. In practice, we replace expectation with average over mini-batch samples. For CNNs a mini-batch has dimensions $(B, C, H, W)$, where we denote B– batch size, C– channels, H– height, W– width. In this case, we reshape this tensor to take the shape $(B \times H \times W, C)$ and treat each row as a hidden vector $\mathbf{h}$.

### 2.1 THEORETICAL ANALYSIS

We now theoretically study the behavior of IB principle on two synthetic datasets designed to provide insights into the invariant representation that IB helps in learning, and simultaneously reveals why it should not be surprising that models trained using maximum likelihood (without appropriate regularization) perform poorly under input perturbations and distribution shift during test time. Although these analyses are done for linear regression, in each case, we empirically verify these predictions on deep ReLU networks. For our analysis, we use the following objective,

$$J(\theta) = \mathbb{E}[(f_\theta(\mathbf{x}) - y)^2] + \lambda\|\theta\|^2 + \frac{\beta}{2\pi e}e^{H(f_\theta(\mathbf{x})|y)} \tag{8}$$

Here the IB regularization $H(f_\theta(\mathbf{x})|y)$ is kept in the exponent for the ease of analytical simplicity. Also, setting $\beta = 0$ yields our baseline case without the IB regularization.

### 2.1.1 Synthetic Dataset A

Minimizing the class conditional entropy forces the distribution of neural network representation corresponding to each class to have the minimum amount of information about the input data. Therefore combining this regularization with the traditional classification losses (Eg. cross-entropy) should encourage the neural network to pick features that are dominantly present in class samples and able to discriminate between samples from different classes. To formalize the above intuition, we consider the following synthetic data generating process where the data samples $\mathbf{x} \in \mathbb{R}^d$ and labels $y$ are sampled from the following distribution,

$$y \sim \{-1, 1\} \qquad x_i \sim \begin{cases} \mathcal{N}(y, \sigma^2) & \text{with probability } p_i \\ \mathcal{N}(-y, \sigma^2) & \text{with probability } 1 - p_i \end{cases} \tag{9}$$

where $i \in \{1, 2, \cdots, d\}$, $y$ is drawn with uniform probability, and $\mathbf{x} = [x_1, x_2, \cdots, x_d]$ is a data sample. Also, all $x_i|y$ are independent of each other. Thus depending on the value of $p_i$, a feature $x_i$ has a small or large amount of information about the label $y$. Specifically, values of $p_i$ close to 0.5 do not tell us anything about the value of $y$ while values close to 0 and 1 can reliably predict its value. Here we make the assumption that features with $p_i$ closer to 0.5 are non-robust features whose distribution may shift during test time, while features with $p_i$ closer to 0 and 1 are robust ones. Thus we would ideally want a trained model to be insensitive to non-robust features. The theorem below shows how the model parameters depend on input dimensions for the optimal parameters when training a linear regression model $f_\theta(\mathbf{x}) := \theta^T \mathbf{x}$ using the IB objective.

**Theorem 1** *Let $\theta^*$ be the minimizer of $J(\theta)$ in Eq. 8 where we have used synthetic dataset A. Then for a large enough $d$, $\theta^* = \mathbf{M}^{-1}|2\mathbf{p} - \mathbf{1}|$, where $\mathbf{M} := \mathbf{\Sigma} + \lambda \mathbf{I} + \beta(\sigma^2 \mathbf{I} + 4 diag(\mathbf{p} \odot (\mathbf{1} - \mathbf{p})))$, such that $\mathbf{\Sigma}$ is a positive definite matrix if[1] $p_i \notin \{0, 0.5, 1\}$ for all $i$.*

As an implication of the above statement, since $\mathbf{M}^{-1}$ is a full rank matrix, aside from the effects due to $\mathbf{\Sigma}$ (which is data dependent and beyond our control), $\theta_i^*$ can in general be non-zero for all input and output correlations. This is especially the case when $\beta = 0$ (no IB regularization). When using a sufficiently large $\beta$, we find that $\theta_i^*$ gets reduced for larger values of $p_i(1 - p_i)$, i.e., when $p_i$ is closer to 0.5. Thus the IB objective can help suppress dependence of the learned model on non-robust (low correlation) features. Although this analysis is for linear regression, it provides evidence that it should not be surprising that deep networks trained with maximum likelihood objective without an appropriate regularization could exhibit a similar behavior. Note that this is not a problem when training and test set are sampled IID from the same distribution, but only becomes one under distribution shift. Also note that since the analysis depends on expectation over data distribution, a larger training set cannot solve our problem of avoiding dependence on non-robust features.

To verify that the behavior studied above also holds for deep networks, we conduct experiments with both linear and deep models on samples drawn randomly from synthetic dataset A. Details of the experimental setup can be found in appendix B.

In figure 1 (left), we plot the parameters $\theta_i^*$ vs. $p_i$ for the linear regression model. Since the same analysis cannot be done for deep networks, we use the perspective that the output-input sensitivity $\mathbf{s}^*$, where $s_i^* := \mathbb{E}_{\mathbf{x}} \left[ \left| \left| \frac{\partial f_{\theta^*}(\mathbf{x})}{\partial x_i} \right| \right| \right]$, is equal to $\theta_i^*$ for linear regression. So for deep networks, we plot $s_i^*$ vs. $p_i$ instead as shown in figure 1 (right). In both models, we normalize the sensitivity values so that the maximum value is 1 for the ease of comparison across different $\beta$ values. Both for linear and deep models, we find that the sensitivity profile goes to 0 away from $p_i = 0$ and 1 when applying the IB regularization with larger values of coefficients $\beta$; this effect being more dramatic for deep networks. Thus the IB regularization helps suppress the dependence of model on non-robust (low correlation) features whose distribution may shift during test time, thus allowing its predictions to be invariant to such shifts.

Here we additionally note that for a linear regression model, while sensitivity is same as the model parameter $\theta$, it is not merely the first-order sensitivity that gets suppressed for certain input dimensions, the output becomes invariant to such dimensions altogether. Although this argument does not necessarily apply to deep networks, note that the IB regularization itself enforces a more gen-

---

[1]This assumption is needed due to technicality.

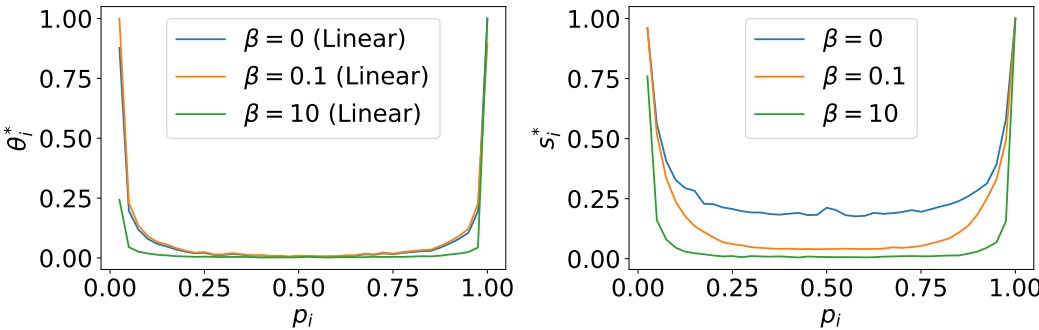

Figure 1: Sensitivity $s_i^*$ of output $f_{\theta^*}(\mathbf{x})$ with respect to input dimensions $x_i$ vs. the probability $p_i$ (controlling correlation between input dimension $i$ and target) for synthetic dataset A (Eq. 9). Left plot shows $\theta_i^*$ (same as sensitivity) computed for a trained linear model. Right plot shows sensitivity computed for a trained MLP. IB regularization acts as a filter, suppressing the sensitivity of both these models to weak correlation features ($p_i$ close to 0.5).

eral condition of finding low entropy representation rather than merely suppressing input sensitivity. Hence the implications of IB could be more general than what our sensitivity analysis shows.

### 2.1.2 SYNTHETIC DATASET B

Using the same intuition that small class conditional entropy induces learned representations to have less uncertainty, given two features that can equally differentiate between classes in expectation, the IB objective should pick the one with smaller variance. To formalize this intuition, we consider the following binary classification problem where the data samples $\mathbf{x} \in \mathbb{R}^d$ and labels $y$ are sampled from the following distribution,

$$y \sim \{-1, 1\} \qquad x_i \sim \begin{cases} \mathcal{N}(y, \sigma^2) & \text{with probability } p_i \\ \mathcal{N}(y, k\sigma^2) & \text{with probability } 1 - p_i \end{cases} \qquad (10)$$

where $i \in \{1, 2, \cdots, d\}$, $y$ is drawn with uniform probability, and $\mathbf{x} = [x_1, x_2, \cdots, x_d]$ is a data sample. Once again, all $x_i|y$ are independent of each other. Thus depending on the value of $p_i$ and $k$, a feature $x_i$ has a small or large variance. We would ideally like the model to avoid dependence on dimensions with high variance because they are non-robust and a minor shift in their distribution during test time can affect the model's decision by a large amount. The theorem below shows how the model parameters depend on input dimensions for the optimal parameters when training a linear regression model $f_\theta(\mathbf{x}) := \theta^T \mathbf{x}$ using the IB objective.

**Theorem 2** *Let $\theta^*$ be the minimizer of $J(\theta)$ in Eq. 8 where we have used synthetic dataset B. Then for a large enough $d$, $\theta^* = \mathbf{M}^{-1}\mathbf{1}$, where, $\mathbf{M} := \mathbf{\Sigma} + \lambda\mathbf{I} + \beta\sigma^2 diag(\mathbf{p} + k(\mathbf{1} - \mathbf{p}))$, such that $\mathbf{\Sigma}$ is a positive definite matrix.*

Once again, we find that $\theta_i^*$ is non-zero for all dimensions of the input. Assume without loss of generality that $k > 1$. Then using a sufficiently large $\beta$ would make the value of $\theta_i^*$ approach 0 if $p_i$ is close to 0. In other words, IB regularization forces the model to be less sensitive to features with high variance. Thus, such a model's prediction will not be affected significantly under a shift of the distribution of high variance features during test time.

To study the extent of similarity of this behavior between linear regression and deep networks, we once again conduct experiments with both these models on a finite number of samples drawn randomly from synthetic dataset B with $k = 10$ and $\sigma^2 = 0.001$. The rest of the details regarding dataset generation and models and optimization are identical to what was used in section 2.1.1.

The sensitivity $s_i^*$ vs. $p_i$ plots are shown in figure 2 (left) for linear regression and figure 2 (right) for MLP. In the case of linear regression $s_i^* = \theta_i^*$. For both linear regression and MLP, the model's sensitivity to all features are high irrespective of $p_i$ when trained without the IB regularization ($\beta = 0$) and this is especially more so for the MLP. On the other hand, when training with IB regularization, we find that a larger $\beta$ forces the models to be less sensitive to input feature dimensions with higher variance (which correspond to $p_i = 0$). The discussion around the generality of the IB regularization beyond sensitivity analysis is same as that discussed in section 2.1.1.

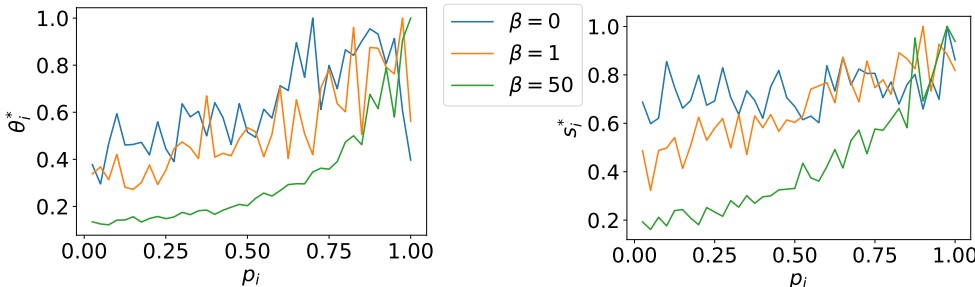

Figure 2: Sensitivity $s_i^*$ of output $f_{\theta^*}(\mathbf{x})$ with respect to input dimensions $x_i$ vs. the probability $p_i$ (deciding the choice between feature with variance $\sigma^2$ vs. $10\sigma^2$) for synthetic dataset B (Eq. 10). Left plot shows $\theta_i^*$ (same as sensitivity) computed for a trained linear model. Right plot shows sensitivity computed for a trained MLP. IB regularization suppresses the sensitivity of both these models to large variance features ($p_i$ close to 0).

## 3    EXPERIMENTS WITH DATA DISTRIBUTION SHIFT

The experiments below are aimed at investigating: 1. the ability of relevant existing methods to generalize under distribution shift; 2. how well can the proposed method generalize under this shift. Details not mentioned in the main text can be found in appendix B.

**Datasets**: We use a colored version of the MNIST dataset (see appendix A for dataset samples and details) for experiment 1, and MNIST-M (Ganin et al., 2016), SVHN (Netzer et al., 2011), MNIST (LeCun & Cortes, 2010) in addition to C-MNIST for experiment 2. All image pixels lie in 0-1 range and are not normalized. The reason for this is that since we are interested in out of distribution (OOD) classification, the normalization constants of training distribution and OOD may be different, in which case data normalized with different statistics cannot be handled by the same network easily.

**Other Details**: We use ResNet-56 (He et al., 2016b) in all our experiments. We use Adam optimizer (Kingma & Ba, 2014) with batch size 128 and weight decay 0.0001 for all experiments unless specified otherwise. We do not use batch normalization in any experiment except for the adaptive batch normalization baseline method. Discussion and experiments around batch normalization can be found in appendix D. We do not use any bias parameter in the network because we found it led to less overfitting overall. For all configurations specified for proposed method and baseline methods below, the hyper-parameter learning rate was chosen from $\{0.0001, 0.001\}$ unless specified otherwise. For entropy penalty, the regularization coefficient is chosen from $\{0.1, 1, 10\}$.

**Baseline methods**:

1. Vanilla maximum likelihood (MLE) training: Since there are no regularization coefficients in this case, we search over batch sizes from $\{32, 64, 128\}$ for each learning rate value.

2. Variational bottleneck method (VIB, Alemi et al. (2016)) is an existing approximation to the IB objective that uses a non-deterministic network. We therefore investigate its behavior under distribution shift at test time. The regularization coefficient for VIB is chosen from the set $\{0.01, 0.1, 1, 5\}$.

3. Clean logit pairing (CLP): Proposed in Kannan et al. (2018), this method minimizes the $\ell^2$ norm of the difference between the logits of different samples. As shown in proposition 3 (in appendix), minimizing this $\ell^2$ norm is equivalent to minimizing the entropy of the distribution in logit space under the assumption that this distribution is Gaussian. In contrast entropy penalty minimizes the entropy of the class conditional distribution of the first hidden layer. Due to this similarity, we consider CLP a baseline. The regularization coefficient for CLP is chosen from $\{0.1, 0.5, 1, 10\}$.

4. Projected gradient descent (PGD) based adversarial training (Madry et al., 2017) has been shown to yield human interpretable features. This makes it a good candidate for investigation. For PGD, $\ell_{\inf}$ perturbation is used with a maximum perturbation $\epsilon$ from the set $\{8, 12, 16, 20\}$ and step size of 2, where all these numbers are divided by 255 since the input is normalized to lie in $[0, 1]$ . The number of PGD steps is chosen from the set $\{20, 50\}$. We randomly choose 12 different configurations out of these combinations.

5. Adversarial logit pairing (ALP, Kannan et al. (2018)) is another approach for adversarial robustness and an alternative to PGD. Since it has the most number of hyper-parameters, we tried a larger number of configurations for this baseline. Specifically, we use $\ell_{\inf}$ norm with a maximum

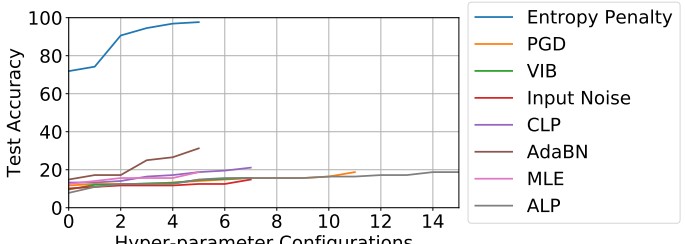

Table 1: Out of distribution performance on test sets using a model trained with Entropy Penalty on C-MNIST dataset.

| Dataset | Accuracy |
|---------|----------|
| C-MNIST | 96.88 |
| MNIST | 93.75 |
| MNIST-M | 85.94 |
| SVHN | 60.94 |

Figure 3: Performance on the distribution shifted test set of C-MNIST for various methods trained on C-MNIST training set. See figure 5 in appendix for samples from C-MNIST dataset.

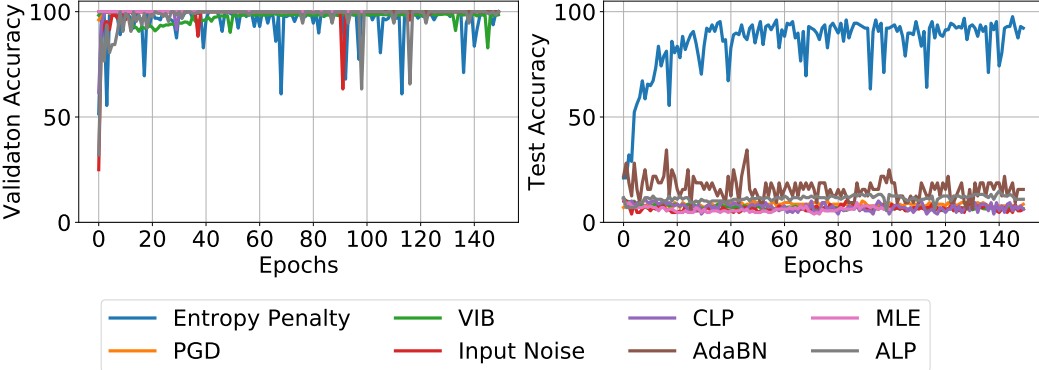

Figure 4: Baseline methods severely overfit color features in the C-MNIST training set leading to near $100\%$ accuracy on C-MNIST validation set but close to chance performance on the distribution shifted C-MNIST test set.

perturbation $\epsilon$ from the set $\{8, 16, 20\}$ and step size of 2, where all these numbers are divided by 255 since the input is normalized to lie in $[0, 1]$. The number of PGD steps is chosen from the set $\{20, 50\}$. The regularization coefficient is chosen from $\{0.1, 1, 10\}$. We randomly choose 15 different configurations out of these combinations.

6. Gaussian Input Noise has been shown to have a similar effect as that from adversarial training (Ford et al., 2019) with even better performance in certain cases. We choose Gaussian input noise with standard deviation from the set $\{0.05, 0.1, 0.2, 0.3\}$.

7. Adaptive batch normalization (AdaBN, Li et al. (2016)) has been proposed as a simple way to achieve domain adaptation in which the running statistics of batch normalization are updated with the statistics of the target domain data. Although this does not fall within our goal of learning a model that does not need any adaption during test time, we investigate this method in experiment 1 due to its simplicity. Since there are no regularization coefficients in this case, we search over batch sizes from $\{32, 64, 128\}$ for each learning rate value.

**Experiment 1**: In this experiment, we train ResNet-56 on the colored MNIST dataset using the baseline methods and entropy regularization, and test the performance of the trained models on the distribution shifted test set of colored MNIST dataset in each case. For each method, we record the best test performance for each hyper-parameter configuration used, and after sorting these numbers across all configurations, we plot them in figure 3. We find that all the baseline methods tend to severely overfit the non-robust color features in C-MNIST leading to poor performance on the distribution shifted test set of C-MNIST. Figure 4 further confirms this by plotting the validation and test accuracy vs. epoch for all methods for one of the hyper-parameter configurations (see appendix C for details). Clearly, baseline methods achieve near $100\%$ accuracy on C-MNIST validation set but close to chance performance on the distribution shifted C-MNIST test set, showing that these methods have overfitted the color features. Entropy penalty is able to avoid this dependence.

Surprisingly even VIB suffers from this issue. This could be because of improper minimization of the information bottleneck (IB) regularization, which could in turn be due to 1. the same reason due to which entropy penalty does not work when applied to higher layers; 2. VIB minimizes an upper bound of the original IB objective. Entropy penalty is able to overcome these difficulties.

**Experiments 2**: In this experiment, we hand-pick the model trained with entropy penalty on C-MNIST in experiment 1 above, such that it simultaneously performs well on SVHN, MNIST-M and MNIST datasets (see section 5 for discussion on this). We used the C-MNIST test set for early stopping. These performances are shown in table 1. We note that it is non-trivial for a single model to perform well on all datasets with such distribution shifts without any domain adaptation, especially given it is trained on a dataset on which all baseline methods severely overfit to non-robust features.

## 4 RELATED WORK

**Invariant Risk Minimization**: The goal of IRM (Arjovsky et al., 2019) is to achieve out of distribution generalization by learning representations such that there exists a predictor (Eg. a linear classifier) that is simultaneously optimal across all environments. IRM achieves this goal by learning (stable) features whose correlation with target is invariant across different environments. In other words, if there are multiple features that correlate with label, then IRM aims to learn the feature which has the same degree of correlation with label irrespective of the environment, while ignoring other features. If the representation induced by such stable features, among others, simultaneously also contain the minimum amount of information about the input, then such representations can alternatively be learned using the information bottleneck principle. Thus it boils down to which strategy forms a better inductive bias for handling distribution shift. On a practical note, the main difference between IRM and our proposal is that IRM requires the explicit knowledge of the environment from which each training data is sampled from. Our approach does not have this restriction. Due to this, we cannot evaluate IRM in our experimental setting.

**Adversarial Training**: There is an abundance of literature around robust optimization (Wald, 1945; Ben-Tal et al., 2009) and adversarial training (Goodfellow et al., 2014; Madry et al., 2017) which study robustness of models to small perturbations around input samples and are often studied using first order methods. Such perturbations can be seen as the worst case distribution shift in the local proximity of the original training distribution. Further, Tsipras et al. (2018) discusses that the representations learned by adversarially trained deep network are more human interpretable. These factors make it a good candidate for investigating its behavior under distribution shift.

Our theoretical analysis has similarities to this line of work, but our goal and conclusions are broader. Specifically, for linear regression, we derive the optimal parameter value analytically under the information bottleneck objective. Since, the value of parameters is same as the output-input sensitivity– the derivative of this model's output (not loss) with respect to its input, we plot sensitivity in the case of deep networks because parameters do not correspond to input dimensions for deep networks. Nonetheless, this is a limitation of our analysis and not of the information bottleneck principle because its objective of minimizing representation entropy is more general than reducing first order sensitivity of the model.

**Domain Adaptation**: Domain adaptation (Wang & Deng, 2018; Patel et al., 2014) addresses the problem of distribution shift between source and target domain, and has attracted considerable attention in computer vision, NLP and speech communities (Kulis et al., 2011; Blitzer et al., 2007; Hosseini-Asl et al., 2018). Some of these methods address this issue by aligning the two distributions (Jiang & Zhai, 2007; Bruzzone & Marconcini, 2009), while others by making use of adversarial training (Ganin & Lempitsky, 2014; Ganin et al., 2016) and auxilliary losses (Ghifary et al., 2015; He et al., 2016a). A common characteristic of all these methods is that they require labeled/unlabeled target domain data during the training process. This is not necessary in the information bottleneck approach, which makes it more flexible.

## 5 DISCUSSION AND CONCLUSION

Based on our analysis, it appears that deep networks are good at achieving state-of-the-art generalization in common settings because a. they are able to exploit all the correlations present between inputs and targets; and b. the IID assumption holds between training and test sets. However, these attributes also make them perform poorly on distribution shifted test sets. Our analysis provides evidence that the information bottleneck (IB) principle can be a potential remedy to this problem. We reached this conclusion by introducing entropy penalty– an equivalent form of the IB regularization for deterministic networks, and showing it generalizes well on out of distribution test sets.

However, note that while entropy penalty itself is a general form of regularization, our proposed implementation of entropy penalty has certain limitations and it lacks of complete theoretical understanding. Specifically,

1. The Gaussian distribution assumption of hidden representation is a limitation and may not apply to more general datasets other than MNIST, where class samples have multi-modal features. This requires alternate ways of minimizing the entropy of distributions which is currently a hard open problem. Additionally, since entropy penalty works best when there is a significant gap between correlation/variance of robust and non-robust features (see section 2.1), it may not be easy to get good OOD performance when training set does not have this property. As evidence, we found this to be the case when training with entropy penalty on SVHN and testing on other datasets (not shown). Two possible solutions to this problem could be: a. design more generic algorithms for minimizing entropy of hidden representation; b. data augmentation techniques that selectively amplify the difference in levels of correlation of robust feature with the target vs. the non-robust ones.

2. Despite our attempt to explain why entropy penalty improves performance under distribution shift when applied to the first hidden layer of deep networks, but not higher layers, a more detailed understanding of this phenomenon remains elusive and is left as future work.

Disjoint from above discussion, the traditional practice of using a validation set for early stopping and selecting hyper-parameters is based on the assumption that training/validation/test sets are sampled IID from the same distribution (Arlot et al., 2010). However, it is not clear how to early stop and select hyper-parameter values when the goal is to evaluate on out of distribution test sets. This is because a set of non-robust features can be shared between a training and validation set, and thus a high performance on such a validation set does not necessarily imply the learned model can generalize to out of distribution test sets. This topic needs further attention.

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

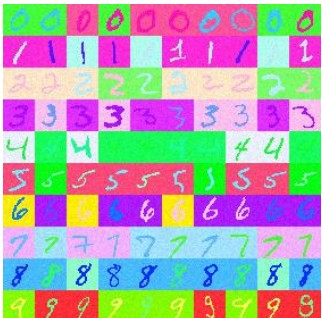 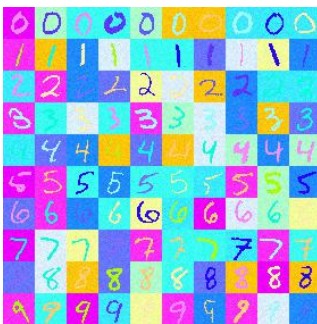

Figure 5: Color MNIST training set (left) and out of distribution test set (right). Each class in training set has two background colors and two foreground colors that are unique only to that class. Each test image has a foreground and background color that is randomly picked out of 10 colors that are chosen independently of the training set.

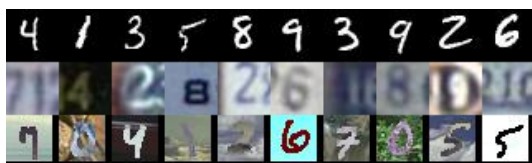

Figure 6: Random samples from MNIST (top), SVHN (middle) and MNIST-M (bottom) datasets are shown to get a visual sense of the hardness of the out of distribution task.

APPENDIX

# A DATASETS

**Colored MNIST Dataset**: Randomly drawn training and test samples from the C-MNIST dataset generated as described above are shown in figure 5. The colored MNIST dataset (C-MNIST), which is used in experiments 1 and 2, uses all the 60,000 training image in the MNIST dataset to generate the C-MNIST dataset, where we randomly do a 0.9-0.1 split to get the training set and validation set respectively. Similarly, we use all the 10,000 test images in MNIST to generate the C-MNIST test set. We vary both the foreground and background colors to generate our C-MNIST dataset. The reason why we vary colors of both foreground and background is that we want the trained model to avoid overfitting any color bias. A single color in foreground or background would constitute a low variance feature, which as we study in section 2.1.2, leads models to prioritize learning it for both training with vanilla MLE as well as MLE with entropy penalty.

The C-MNIST training/validation set is generated from its MNIST counter-part as follows:

1. For each class, randomly assign two colors (RGB value) for foreground and two colors (RGB value) for background.

2. Binarize each image (pixels in 0-255 range) around threshold 150 so that pixel values are either 0 or 1 and replicate the channel in each image to have a three channel image.

3. For each image in a class, randomly pick one of the two foreground colors assigned to that class and replace all foreground pixel with that color. Similarly replace background pixels for all images.

4. Add zero mean Gaussian noise with a small standard deviation (0.04 used in our experiments) to all images.

To generate the test set, in step one, we randomly assign a foreground and background color to each image irrespective of the class, and the colors for validation set are chosen independently of the training set.

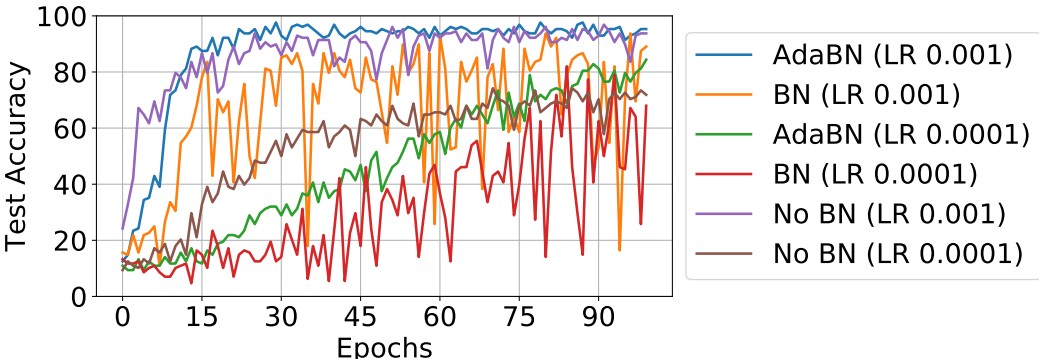

Figure 7: Batch Normalization is unstable without AdaBN when using Entropy Penalty (experiment on C-MNIST dataset).

**Other Datasets**: During experiments with MNIST, the single channel in each image was replicated to form a three channel image. For SVHN, we resized the image to have $28 \times 28$ hieght and width.

## B  EXPERIMENTAL DETAILS

**Experiment in Section 2.1.1**: The input data is in 500 dimensions and set the value of $p_i$ for each $i$ to be uniformly from $[0, 1]$ and fix it henceforth. We then randomly sample 15000 input-target pairs from this dataset with $\sigma^2 = 0.0001$. We train a linear regression model and a 3 hidden layer perceptron (MLP) of width 200 with ReLU activation for 1000 iterations using Adam optimizer with learning rate 0.001 and weight decay 0.00001.

**Variational bottleneck method** (VIB): As an implementation detail of VIB, we flatten the output of the last convolution layer of ResNet-56 and separately pass it through two linear layer of width 256, one that outputs a vector that we regard as mean $\mu$, and the other that we regard as log-variance $\nu$ (similar to how it is done in variational auto-encoders (Kingma & Welling, 2013)). We then combine their outputs as $\mathbf{o} = \mu + exp(0.5\nu) \odot \epsilon$ where $\epsilon$ is sampled from a standard Gaussian distribution of the same dimension as $\nu$. This output is then passed through a linear layer that transforms it into a vector of dimension same as the number of classes. These implementation details are similar to those in the original VIB paper.

Further, we gradually ramp up the regularization coefficient $\beta$ from 0.0001 to its final value by doubling the current value at the end of every epoch until the final value is reached. This is a popular prctice when minimizing the KL divergence term between posterior and prior which helps optimization.

## C  OVERFITTING COLOR FEATURES

The validation set of C-MNIST is a held out set that follows the same distribution as the training set but the two are mutually exclusive. On the other hand, the test set of C-MNIST has different foreground and background colors in all images that are sampled independently of the training set as explained in section A and shown in figure 5. The hyper-parameter configuration chosen (from among the ones used in experiment 1 in section 3) for each method is based on the condition that training accuracy converged to $100\%$ at the end of the training process. This ensures a fair comparison of validation and test accuracy between different methods.

## D  EFFECTS OF BATCH NORMALIZATION ON ENTROPY PENALTY

In this section we study the effect of batch normalization (BN, Ioffe & Szegedy (2015)) when using it in conjunction with entropy penalty. We train a ResNet-56 on C-MNIST dataset identical to the settings in experiment 1 in main text, with the exception that we use BN. While doing so, we

record the accuracy of the model on the C-MNIST test set at every epoch. In addition, we also run experiments where under the same training settings, during evaluation at each epoch, we use AdaBN (Li et al., 2016). These values are plotted in figure 7. We have also plotted runs using entropy penalty without any batch normalization for reference. In all cases, we use a learning rates (LR) of 0.001 and 0.0001. The plots show that without AdaBN, test accuracy is very unstable on the distribution shifted test set. This seems to be a side-effect of using BN which can be fixed by adaptive the running statistics of BN (used during evaluation) with the test set statistics. However, this requires domain knowledge (i.e., samples) of the test set, which is not preferable for our goal.

## E  PROOFS

**Proposition 2** $\hat{\mathcal{L}}_{IB}(\theta) = (1 - \beta)\mathcal{H}(\mathbf{Y}|f_\theta(\mathbf{X})) + \beta\mathcal{H}(f_\theta(\mathbf{X})|\mathbf{Y}) + C$

*where $C$ is a positive constant for discrete $\mathbf{Y}$, independent of $\theta$.*

**Proof**: *We note that,*

$$\mathcal{H}(f_\theta(\mathbf{X})) = -\int \Pr(f_\theta(\mathbf{X})) \log \Pr(f_\theta(\mathbf{X})) \tag{11}$$

$$= -\sum_{\mathbf{Y}} \int \Pr(f_\theta(\mathbf{X}), \mathbf{Y}) \log \Pr(f_\theta(\mathbf{X})) \tag{12}$$

$$= -\sum_{\mathbf{Y}} \int \Pr(f_\theta(\mathbf{X}), \mathbf{Y}) \left( \log \Pr(f_\theta(\mathbf{X})|\mathbf{Y}) + \log \frac{\Pr(f_\theta(\mathbf{X}))}{\Pr(f_\theta(\mathbf{X})|\mathbf{Y})} \right) \tag{13}$$

$$= -\sum_{\mathbf{Y}} \int \Pr(f_\theta(\mathbf{X}), \mathbf{Y}) \left( \log \Pr(f_\theta(\mathbf{X})|\mathbf{Y}) - \log \Pr(\mathbf{Y}|f_\theta(\mathbf{X})) + \log \Pr(\mathbf{Y}) \right) \tag{14}$$

$$= \mathcal{H}(f_\theta(\mathbf{X})|\mathbf{Y}) - \mathcal{H}(\mathbf{Y}|f_\theta(\mathbf{X})) + \mathcal{H}(\mathbf{Y}) \tag{15}$$

*where the fourth equality is due to Bayes rule. Since $\mathbf{Y}$ is discrete, $\mathcal{H}(\mathbf{Y})$ is a positive constant. Finally, substituting the above expression in the definition of $\hat{\mathcal{L}}_{IB}(\theta)$, yields,*

$$\hat{\mathcal{L}}_{IB}(\theta) = \mathcal{H}(\mathbf{Y}|f_\theta(\mathbf{X})) + \beta(\mathcal{H}(f_\theta(\mathbf{X})|\mathbf{Y}) - \mathcal{H}(\mathbf{Y}|f_\theta(\mathbf{X})) + \mathcal{H}(\mathbf{Y})) \tag{16}$$

$$= (1 - \beta)\mathcal{H}(\mathbf{Y}|f_\theta(\mathbf{X})) + \beta\mathcal{H}(f_\theta(\mathbf{X})|\mathbf{Y}) + \beta\mathcal{H}(\mathbf{Y}) \tag{17}$$

*which proves the claim.* $\square$

**Lemma 1**

$$\mathbb{E}[x_i|y = 1] = -\mathbb{E}[x_i|y = -1] = 2p_i - 1 \tag{18}$$

$$\mathbb{E}[x_i^2|y = 1] = \mathbb{E}[x_i^2|y = -1] = 1 + \sigma^2 \tag{19}$$

**Proof**: *Given the distribution of $\mathbf{x}$, we can write each element $x_i$ as,*

$$x_i = b_i n_i + (1 - b_i)\bar{n}_i \tag{20}$$

*where $b_i$ is sampled from the Bernoilli distribution with probability $p_i$, $n_i \sim \mathcal{N}(y, \sigma^2)$, and $\bar{n}_i \sim \mathcal{N}(-y, \sigma^2)$. Thus,*

$$\mathbb{E}[x_i|y = 1] = p_i - (1 - p_i) = 2p_i - 1 \tag{21}$$

$$\mathbb{E}[x_i|y = -1] = -p_i + (1 - p_i) = 1 - 2p_i \tag{22}$$

*Next,*

$$\mathbb{E}[x_i^2|y = 1] = \mathbb{E}[b_i^2 n_i^2 + (1 - b_i)^2 \bar{n}_i^2 + 2b_i(1 - b_i)n_i\bar{n}_i|y = 1] \tag{23}$$

*We know from the properties of Bernoulli and Gaussian distribution that,*

$$\mathbb{E}[b_i^2] = var(b_i) + \mathbb{E}[b_i]^2 = p_i(1 - p_i) + p_i^2 = p_i \tag{24}$$

$$\mathbb{E}[n_i^2|y = 1] = var(n_i|y = 1) + \mathbb{E}[n_i|y = 1]^2 = \sigma^2 + 1 \tag{25}$$

*We similarly get,*

$$\mathbb{E}[n_i^2|y=-1] = \sigma^2 + 1 \tag{26}$$

$$\mathbb{E}[\bar{n}_i^2|y=1] = \sigma^2 + 1 \tag{27}$$

$$\mathbb{E}[\bar{n}_i^2|y=-1] = \sigma^2 + 1 \tag{28}$$

*Therefore, using the independence property between random variables,*

$$\mathbb{E}[x_i^2|y=1] = \mathbb{E}[b_i^2]\mathbb{E}[n_i^2|y=1] + (1 + \mathbb{E}[b_i^2] - 2\mathbb{E}[b_i])\mathbb{E}[\bar{n}_i^2|y=1]$$

$$+ 2(\mathbb{E}[b_i] - \mathbb{E}[b_i]^2)\mathbb{E}[\bar{n}_i|y=1]\mathbb{E}[n_i|y=1] \tag{29}$$

$$= 1 + \sigma^2 \tag{30}$$

*We similarly have,*

$$\mathbb{E}[x_i^2|y=-1] = 1 + \sigma^2 \tag{31}$$

**Theorem 3** *Let $\theta^*$ be the minimizer of $J(\theta)$ in Eq. 8 where we have used synthetic dataset A. Then for a large enough $d$, $\theta^*$ is given by,*

$$\theta^* = \mathbf{M}^{-1}|2\mathbf{p} - \mathbf{1}| \tag{32}$$

*where,*

$$\mathbf{M} := \mathbf{\Sigma} + \lambda\mathbf{I} + \beta(\sigma^2\mathbf{I} + 4diag(\mathbf{p} \odot (\mathbf{1} - \mathbf{p}))) \tag{33}$$

*such that $\mathbf{\Sigma}$ is a positive definite matrix if[2] $p_i \notin \{0, 0.5, 1\}$ for all $i$.*

***Proof:*** *We start by noting that,*

$$H(f_\theta(\mathbf{x})|y) = -\sum_{y\in\{-1,1\}}\int_{\mathbf{x}} \Pr(f_\theta(\mathbf{x}), y)\log\Pr(f_\theta(\mathbf{x})|y) \tag{34}$$

$$= -0.5\int_{\mathbf{x}} \Pr(f_\theta(\mathbf{x})|y=1)\log\Pr(f_\theta(\mathbf{x}|y=1)) + \Pr(f_\theta(\mathbf{x})|y=-1)\log\Pr(f_\theta(\mathbf{x}|y=-1)) \tag{35}$$

*Identifying that,*

$$H_{f_\theta(\mathbf{x})|y=1} := -\int_{\mathbf{x}} \Pr(f_\theta(\mathbf{x})|y=1)\log\Pr(f_\theta(\mathbf{x}|y=1)) \tag{36}$$

$$H_{f_\theta(\mathbf{x})|y=-1} := -\int_{\mathbf{x}} \Pr(f_\theta(\mathbf{x})|y=-1)\log\Pr(f_\theta(\mathbf{x}|y=-1)) \tag{37}$$

*are the entropy of the class conditional distributions $\Pr(f_\theta(\mathbf{x})|y=1)$ and $\Pr(f_\theta(\mathbf{x})|y=-1)$ respectively, we have that,*

$$H(f_\theta(\mathbf{x})|y) = 0.5\left(H_{f_\theta(\mathbf{x})|y=1} + H_{f_\theta(\mathbf{x})|y=-1}\right) \tag{38}$$

*We note that,*

$$f_\theta(\mathbf{x})|(y=1) = \sum_{i=1}^{d} \theta_i x_i|y=1 \tag{39}$$

$$= \sum_{i=1}^{d} \theta_i(b_i(n_i|y=1) + (1-b_i)(\bar{n}_i|y=1)) \tag{40}$$

*For a large enough dimensionality $d$ of $\mathbf{x}$, central limit theorem (CLT) applies to $f_\theta(\mathbf{x})|(y=1)$ and it converges to a Gaussian distribution. Thus the entropy of this distribution is given by*

---

[2]This assumption is needed due to technicality.

$0.5 \log(2\pi e s_1^2)$ *where $s_1^2$ is the variance of $f_\theta(\mathbf{x})|y = 1$. A similar argument applies to $f_\theta(\mathbf{x})|y = -1$, in which case we define $s_{-1}^2$ to be its variance. Thus,*

$$s_1^2 = var(\sum_{i=1}^{d} \theta_i x_i | y = 1) \tag{41}$$

$$= \sum_{i=1}^{d} var(\theta_i x_i | y = 1) \tag{42}$$

$$= \sum_{i=1}^{d} \theta_i^2 var(x_i | y = 1) \tag{43}$$

*where the second equality holds because $x_i$'s are independent of one another. Thus using lemma 1,*

$$s_1^2 = \sum_i \theta_i^2 (1 + \sigma^2 - (2p_i - 1)^2) \tag{44}$$

$$= \sum_i \theta_i^2 (\sigma^2 + 4p_i(1 - p_i)) \tag{45}$$

*We similarly get,*

$$s_{-1}^2 = \sum_i \theta_i^2 (\sigma^2 + 4p_i(1 - p_i)) \tag{46}$$

*Since $s_1^2$ and $s_{-1}^2$ are equal, we denote $s^2 = s_1^2 = s_{-1}^2$. Thus,*

$$H_{f_\theta(\mathbf{x})|y=1} = 0.5 \log(2\pi e s^2) \tag{47}$$
$$H_{f_\theta(\mathbf{x})|y=-1} = 0.5 \log(2\pi e s^2) \tag{48}$$

*and,*

$$H(f_\theta(\mathbf{x})|y) = 0.5(0.5 \log(2\pi e s^2) + 0.5 \log(2\pi e s^2)) \tag{49}$$
$$= 0.5 \log(2\pi e s^2) \tag{50}$$

*Therefore, our objective becomes,*

$$\arg \min_\theta \mathbb{E}[(f_\theta(\mathbf{x}) - y)^2] + \lambda \|\theta\|^2 + \beta s^2 \tag{51}$$

$$= \arg \min_\theta \theta^T \mathbb{E}[\mathbf{x}\mathbf{x}^T]\theta - 2\theta^T \mathbb{E}[\mathbf{x}y] + \lambda \|\theta\|^2 + \beta \sum_{i=1}^{d} \theta_i^2(\sigma^2 + 4p_i(1 - p_i)) \tag{52}$$

*Define $\mathbf{M}$ as,*

$$\mathbf{M} := \Sigma + \lambda \mathbf{I} + \beta(\sigma^2 \mathbf{I} + 4 diag(\mathbf{p} \odot (\mathbf{1} - \mathbf{p}))) \tag{53}$$

*where $\Sigma := \mathbb{E}[\mathbf{x}\mathbf{x}^T]$, we can re-write our objective as,*

$$\arg \min_\theta \theta^T \mathbf{M}\theta - 2\theta^T \mathbb{E}[\mathbf{x}y] \tag{54}$$

*whose solution is given by,*

$$\theta^* = \mathbf{M}^{-1} \mathbb{E}[\mathbf{x}y] \tag{55}$$

*Using lemma 1, we get,*

$$\mathbb{E}[x_i y] = \mathbb{E}[x_i | y = 1] \Pr(y = 1) - \mathbb{E}[x_i | y = -1] \Pr(y = -1) \tag{56}$$
$$= 0.5(\mathbb{E}[x_i | y = 1] - \mathbb{E}[x_i | y = -1]) \tag{57}$$
$$= 0.5(4p_i - 2) = 2p_i - 1 \tag{58}$$

*Plugging this value in Eq. 55 yields $\theta^*$.*

*Now we prove that $\Sigma$ is full rank. Note that it is positive semi-definite since it is a scatter matrix. Next, due to conditional independence, for $i \neq j$,*

$$\mathbb{E}[x_i x_j] = \mathbb{E}[x_i x_j | y = 1] \Pr(y = 1) + \mathbb{E}[x_i x_j | y = -1] \Pr(y = -1) \tag{59}$$

$$= \mathbb{E}[x_i | y = 1] \mathbb{E}[x_j | y = 1] \Pr(y = 1) + \mathbb{E}[x_i | y = -1] \mathbb{E}[x_j | y = -1] \Pr(y = -1) \tag{60}$$

*and for $i = j$,*

$$\mathbb{E}[x_i^2] = \mathbb{E}[x_i^2 | y = 1] \Pr(y = 1) + \mathbb{E}[x_i^2 | y = -1] \Pr(y = -1) \tag{61}$$

*Using lemma 1, we get,*

$$\Sigma_{ij} = \begin{cases} 1 + \sigma^2 & \text{if } i = j \\ (1 - 2p_i)(1 - 2p_j) & \text{otherwise} \end{cases} \tag{62}$$

*To prove that $\Sigma$ is positive definite (and hence full rank), we need to prove that no two columns are parallel. To show this, consider any two indices $i$ and $j$ such that $i \neq j$. We show that there exists no $\alpha \neq 0$ such that the columns $\Sigma_i = \alpha \Sigma_j$. We prove this by contradiction. Suppose $\Sigma_{ii} = \alpha \Sigma_{ji}$, then $\alpha \Sigma_{jj} = \frac{\Sigma_{ii} \Sigma_{jj}}{\Sigma_{ji}}$. Substituting values from Eq. 62,*

$$\alpha \Sigma_{jj} = \frac{(1 + \sigma^2)^2}{(1 - 2p_i)(1 - 2p_j)} \tag{63}$$

*Thus $\alpha \Sigma_{jj} > 1$. However, $\Sigma_{ij} = (1 - 2p_i)(1 - 2p_j) < 1$. Thus there is no non-zero $\alpha$ for which $\Sigma_i = \alpha \Sigma_j$. Hence $\Sigma$ must be full rank and hence positive definite. Thus we have proved the claim.* $\square$

**Lemma 2**

$$\mathbb{E}[x_i | y = 1] = -\mathbb{E}[x_i | y = -1] = 1 \tag{64}$$

$$\mathbb{E}[x_i^2 | y = 1] = \mathbb{E}[x_i^2 | y = -1] = 1 + \sigma^2(p_i + k_i(1 - p_i)) \tag{65}$$

**Proof**: *Given the distribution of $\mathbf{x}$, we can write each element $x_i$ as,*

$$x_i = b_i n_i + (1 - b_i) \bar{n}_i \tag{66}$$

*where $b_i$ is sampled from the Bernoilli distribution with probability $p_i$, $n_i \sim \mathcal{N}(y, \sigma^2)$, and $\bar{n}_i \sim \mathcal{N}(y, k_i \sigma^2)$. Thus,*

$$\mathbb{E}[x_i | y = 1] = p_i + (1 - p_i) = 1 \tag{67}$$

$$\mathbb{E}[x_i | y = -1] = -p_i - (1 - p_i) = -1 \tag{68}$$

*Next,*

$$\mathbb{E}[x_i^2 | y = 1] = \mathbb{E}[b_i^2 n_i^2 + (1 - b_i)^2 \bar{n}_i^2 + 2b_i(1 - b_i) n_i \bar{n}_i | y = 1] \tag{69}$$

*We know from the properties of Bernoulli and Gaussian distribution that,*

$$\mathbb{E}[b_i^2] = var(b_i) + \mathbb{E}[b_i]^2 = p_i(1 - p_i) + p_i^2 = p_i \tag{70}$$

$$\mathbb{E}[n_i^2 | y = 1] = var(n_i | y = 1) + \mathbb{E}[n_i | y = 1]^2 = \sigma^2 + 1 \tag{71}$$

*We similarly get,*

$$\mathbb{E}[n_i^2 | y = -1] = \sigma^2 + 1 \tag{72}$$

$$\mathbb{E}[\bar{n}_i^2 | y = 1] = k\sigma^2 + 1 \tag{73}$$

$$\mathbb{E}[\bar{n}_i^2 | y = -1] = k\sigma^2 + 1 \tag{74}$$

*Therefore, using the independence property between random variables,*

$$\mathbb{E}[x_i^2 | y = 1] = \mathbb{E}[b_i^2] \mathbb{E}[n_i^2 | y = 1] + (1 + \mathbb{E}[b_i^2] - 2\mathbb{E}[b_i]) \mathbb{E}[\bar{n}_i^2 | y = 1]$$
$$+ 2(\mathbb{E}[b_i] - \mathbb{E}[b_i]^2) \mathbb{E}[\bar{n}_i | y = 1] \mathbb{E}[n_i | y = 1] \tag{75}$$

$$= p_i(1 + \sigma^2) + (1 - p_i)(1 + k\sigma^2) \tag{76}$$

*We similarly have,*

$$\mathbb{E}[x_i^2 | y = -1] = p_i(1 + \sigma^2) + (1 - p_i)(1 + k\sigma^2) \tag{77}$$

*Rearranging these terms yields the claim.* $\square$

**Theorem 4** *Let $\theta^*$ be the minimizer of $J(\theta)$ in Eq. 8 where we have used synthetic dataset B. Then for a large enough $d$, $\theta^*$ is given by,*

$$\theta^* = \mathbf{M}^{-1}\mathbf{1} \tag{78}$$

*where,*

$$\mathbf{M} := \mathbf{\Sigma} + \lambda\mathbf{I} + \beta\sigma^2 diag(\mathbf{p} + k(\mathbf{1} - \mathbf{p})) \tag{79}$$

*such that $\mathbf{\Sigma}$ is a positive definite matrix.*

**Proof**: *Similar to theorem 1 we have that,*

$$H(f_\theta(\mathbf{x})|y) = 0.5\left(H_{f_\theta(\mathbf{x})|y=1} + H_{f_\theta(\mathbf{x})|y=-1}\right) \tag{80}$$

*and,*

$$f_\theta(\mathbf{x})|(y=1) = \sum_{i=1}^{d} \theta_i b_i(n_i|y=1) + (1-b_i)(\bar{n}_i|y=1) \tag{81}$$

*For a large enough dimensionality $d$ of $\mathbf{x}$, central limit theorem (CLT) applies to $f_\theta(\mathbf{x})|(y=1)$ and it converges to a Gaussian distribution. Thus the entropy of this distribution is given by $0.5\log(2\pi e s_1^2)$ where $s_1^2$ is the variance of $f_\theta(\mathbf{x})|y=1$. A similar argument applies to $f_\theta(\mathbf{x})|y=-1$, in which case we define $s_{-1}^2$ to be its variance. Thus,*

$$s_1^2 = \sum_{i=1}^{d} \theta_i^2 var(x_i|y=1) \tag{82}$$

*Thus using lemma 2,*

$$s_1^2 = \sum_i \theta_i^2(\sigma^2(p_i + k_i(1-p_i))) \tag{83}$$

*We similarly get,*

$$s_{-1}^2 = \sum_i \theta_i^2(\sigma^2(p_i + k_i(1-p_i))) \tag{84}$$

*Since $s_1^2$ and $s_{-1}^2$ are equal, we denote $s^2 = s_1^2 = s_{-1}^2$. Thus,*

$$H_{f_\theta(\mathbf{x})|y=1} = 0.5\log(2\pi e s^2) \tag{85}$$
$$H_{f_\theta(\mathbf{x})|y=-1} = 0.5\log(2\pi e s^2) \tag{86}$$

*and,*

$$H(f_\theta(\mathbf{x})|y) = 0.5(0.5\log(2\pi e s^2) + 0.5\log(2\pi e s^2)) \tag{87}$$
$$= 0.5\log(2\pi e s^2) \tag{88}$$

*Therefore, our objective becomes,*

$$\underset{\theta}{\arg\min}\, \mathbb{E}[(f_\theta(\mathbf{x}) - y)^2] + \lambda\|\theta\|^2 + \beta s^2 \tag{89}$$

$$= \underset{\theta}{\arg\min}\, \theta^T\mathbb{E}[\mathbf{x}\mathbf{x}^T]\theta - 2\theta^T\mathbb{E}[\mathbf{x}y] + \lambda\|\theta\|^2 + \beta\sigma^2\sum_i \theta_i^2(p_i + k_i(1-p_i)) \tag{90}$$

*Define $\mathbf{M}$ as,*

$$\mathbf{M} := \mathbf{\Sigma} + \lambda\mathbf{I} + \beta\sigma^2 diag(\mathbf{p} + k(\mathbf{1} - \mathbf{p})) \tag{91}$$

*where $\mathbf{\Sigma} := \mathbb{E}[\mathbf{x}\mathbf{x}^T]$, we can re-write our objective as,*

$$\underset{\theta}{\arg\min}\, \theta^T\mathbf{M}\theta - 2\theta^T\mathbb{E}[\mathbf{x}y] \tag{92}$$

*whose solution is given by,*

$$\theta^* = \mathbf{M}^{-1}\mathbb{E}[\mathbf{x}y] \tag{93}$$

*Using lemma 2, we get,*

$$\mathbb{E}[x_i y] = \mathbb{E}[x_i | y = 1] \Pr(y = 1) - \mathbb{E}[x_i | y = -1] \Pr(y = -1) \tag{94}$$

$$= 0.5(\mathbb{E}[x_i | y = 1] - \mathbb{E}[x_i | y = -1]) \tag{95}$$

$$= 1 \tag{96}$$

*Plugging this value in Eq. 93 yields $\theta^*$.*

*Now we prove that $\mathbf{\Sigma}$ is full rank. Note that it is positive semi-definite since it is a scatter matrix. Next, due to conditional independence, for $i \neq j$,*

$$\mathbb{E}[x_i x_j] = \mathbb{E}[x_i x_j | y = 1] \Pr(y = 1) + \mathbb{E}[x_i x_j | y = -1] \Pr(y = -1) \tag{97}$$

$$= \mathbb{E}[x_i | y = 1]\mathbb{E}[x_j | y = 1] \Pr(y = 1) + \mathbb{E}[x_i | y = -1]\mathbb{E}[x_j | y = -1] \Pr(y = -1) \tag{98}$$

*and for $i = j$,*

$$\mathbb{E}[x_i^2] = \mathbb{E}[x_i^2 | y = 1] \Pr(y = 1) + \mathbb{E}[x_i^2 | y = -1] \Pr(y = -1) \tag{99}$$

*Using lemma 2, we get,*

$$\Sigma_{ij} = \begin{cases} 1 + \sigma^2(p_i + k_i(1 - p_i)) & \text{if } i = j \\ 1 & \text{otherwise} \end{cases} \tag{100}$$

*To prove that $\mathbf{\Sigma}$ is positive definite (and hence full rank), we need to prove that no two columns are parallel. To show this, consider any two indices $i$ and $j$ such that $i \neq j$. We show that there exists no $\alpha \neq 0$ such that the columns $\Sigma_i = \alpha \Sigma_j$. We prove this by contradiction. Suppose $\Sigma_{ii} = \alpha \Sigma_{ji}$, then $\alpha \Sigma_{jj} = \frac{\Sigma_{ii} \Sigma_{jj}}{\Sigma_{ji}}$. Substituting values from Eq. 100,*

$$\alpha \Sigma_{jj} = (1 + \sigma^2(p_i + k(1 - p_i)))(1 + \sigma^2(p_j + k(1 - p_j))) \tag{101}$$

*Thus $\alpha \Sigma_{jj} > 1$. However, $\Sigma_{ij} = 1$. Thus there is no non-zero $\alpha$ for which $\Sigma_i = \alpha \Sigma_j$. Hence $\mathbf{\Sigma}$ must be full rank and hence positive definite. Thus we have proved the claim.* $\square$

**Proposition 3** *If $\Pr(f_\theta(\mathbf{X}))$ follows a Gaussian distribution, then,*

$$\mathcal{H}(f_\theta(\mathbf{X})) = 0.5 \log(\pi e \mathbb{E}_{\mathbf{X}_1, \mathbf{X}_2 \sim \mathcal{D}(\mathbf{X})}[(f_\theta(\mathbf{X}_1) - f_\theta(\mathbf{X}_2))^2]) \tag{102}$$

*where $\mathbf{X}_1$ and $\mathbf{X}_2$ are IID samples from the data distribution $\mathcal{D}(\mathbf{X})$.*

**Proof:** *We note that,*

$$\mathcal{H}(f_\theta(\mathbf{X})) = -\Pr(f_\theta(\mathbf{X})) \log \Pr(f_\theta(\mathbf{X})) \tag{103}$$

$$= -\Pr(f_\theta(\mathbf{X})) \log \Pr(f_\theta(\mathbf{X})) \tag{104}$$

$$= -(0.5 \log(2\pi e \sigma^2)) \tag{105}$$

*where $\sigma^2$ is the variance of the Gaussian distribution $\Pr(f_\theta(\mathbf{X}))$.*

*Let $\mu := \mathbb{E}_{\mathbf{X}_1 \sim \mathcal{D}(\mathbf{X})}[f_\theta(\mathbf{X}_1)]$. Then note that $\mathbb{E}_{\mathbf{X}_2 \sim \mathcal{D}(\mathbf{X})}[f_\theta(\mathbf{X}_2)] = \mu$ as well since $\mathbf{X}_1$ and $\mathbf{X}_2$ are IID samples. Therefore,*

$$\mathbb{E}_{\mathbf{X}_1, \mathbf{X}_2 \sim \mathcal{D}(\mathbf{X})}[(f_\theta(\mathbf{X}_1) - f_\theta(\mathbf{X}_2))^2] \tag{106}$$

$$= \mathbb{E}_{\mathbf{X}_1, \mathbf{X}_2 \sim \mathcal{D}(\mathbf{X})}[((f_\theta(\mathbf{X}_1) - \mu) - (f_\theta(\mathbf{X}_2) - \mu))^2] \tag{107}$$

$$= \mathbb{E}_{\mathbf{X}_1 \sim \mathcal{D}(\mathbf{X})}[((f_\theta(\mathbf{X}_1) - \mu)^2] + \mathbb{E}_{\mathbf{X}_2 \sim \mathcal{D}(\mathbf{X})}[(f_\theta(\mathbf{X}_2) - \mu))^2]$$

$$- 2\mathbb{E}_{\mathbf{X}_1, \mathbf{X}_2 \sim \mathcal{D}(\mathbf{X})}[((f_\theta(\mathbf{X}_1) - \mu)(f_\theta(\mathbf{X}_2) - \mu)] \tag{108}$$

$$= 2\mathbb{E}_{\mathbf{X} \sim \mathcal{D}(\mathbf{X})}[((f_\theta(\mathbf{X}) - \mu)^2] \tag{109}$$

$$= 2\sigma^2 \tag{110}$$

*Therefore substituting $\mathbb{E}_{\mathbf{X}_1, \mathbf{X}_2 \sim \mathcal{D}(\mathbf{X})}[(f_\theta(\mathbf{X}_1) - f_\theta(\mathbf{X}_2))^2]$ in Eq. 105 yields the claim.* $\square$

