# OpenReview forum: "Entropy Penalty: Towards Generalization Beyond the IID Assumption"
_ICLR.cc/2020/Conference — Reject_

### Official Review · AnonReviewer3 · 2019-10-23
**Official Blind Review #3**

**Rating:** 1

**Review:**

In this work, the author(s) presented a regularization scheme that intends to suppress the identification of spurious features when learning deep representations. Their construction was inspired by the information bottleneck framework. By making Gaussian assumptions on the form of label conditioned feature distribution, the entropy penalty can be efficiently computed in the form of L2 loss, which is easy to implement. My major concerns for this submission are its clarity, novelty, and theoretical depth. The arguments provided are not very convincing and reported experimental results are based on toy-scale datasets. I recommend rejection for this submission, with more detailed comments attached below.

Strength
+ The author(s) are trying to resolve the issue of learning spurious discriminant features for predictive models, which is a trendy topic with a potential impact on the field.
+ There are some interesting discussions in the related work section.

Weakness
- The presentation needs to be much improved. In its current form, the lack of clarity leads to serious confusion. Examples include but not limited to the following:
	- "violates the IID assumption which is the foundation of existing generalization theory". Not sure what this IID assumption means, please briefly/intuitively describe these classical generalization theories.
	- "all the correlations btw inputs and targets."
	- "throws away maximum possible information about the input distribution"
- The author(s) have made a strong statement, quote "it is the second term that regularizes the model representations to become invariant to non-robust features"
- Eqn (1) and Eqn (3) is equivalent, what's the point??? There is no novelty here.
- Prop 1. Modeling the conditional entropy H(f_{\theta}(X)|Y) nonparametrically is not any easier than modeling the marginal entropy H(f_{\theta}(X)). The assumption of a parametric form of f_{\theta}(X) given Y is very strong and needs to be justified (at least experimentally). Although the author(s) are honest about this limitation in the discussion.
- The concept of distribution shift is not formally introduced in the manuscript.
- Eqn (7) implicitly makes a strong prior assumption that the feature distribution condition on the label is isotropic Gaussian. This reminds me of Linear Discriminant Analysis (LDA), which followed from a similar heuristic, and might partly explain the empirical success of this practice (the model is forced to be LDA like, which combats the overfitting via appealing to simpler models). However, I have not found any discussion related to this, which evidence that the author(s) might lack a proper understanding of classical treatments.
- Theoretical analyses of synthetic examples do not lend strong support to this paper.

Questions
# What is the fundamental difference btw the proposed work differs and domain adaption?

Minors:
% Conditional entropy H(f_{\theta}(X)|X) is zero.
% I do not see the point of referencing adversarial robustness literature.

**Experience Assessment:**

I have read many papers in this area.

**Review Assessment: Checking Correctness Of Derivations And Theory:**

I carefully checked the derivations and theory.

**Review Assessment: Checking Correctness Of Experiments:**

I assessed the sensibility of the experiments.

**Review Assessment: Thoroughness In Paper Reading:**

I read the paper at least twice and used my best judgement in assessing the paper.

---

### Official Review · AnonReviewer2 · 2019-10-23
**Official Blind Review #2**

**Rating:** 3

**Review:**

The paper proposes to combat domain shift via the information bottleneck principle, where the representations are encouraged to learn features that have high mutual information with the labels while having low mutual information with the original data. The paper proposes a specific approach to enforcing this information bottleneck, called "entropy penalty", which tries to minimize the L2 distance between the representation at the first layer and the mean representation of the corresponding class.

The paper proceeds to demonstrate theoretically that some variant of the proposed entropy regularization works for synthetic, binary cases in the sense that it decreases the effect of features that are not highly correlated with the label.

The paper finally demonstrates superior performance on synthetic experiments and classifying digits in an out-of-distribution setting where the OOD distribution is changing foreground and background color.

My greatest concern of the proposed approach is the generality of the method. While the information bottleneck is a very general principle, the way it is implemented in this paper is very specific, which only works for the first hidden layer representation of an input x (even before non-linearity). Combining this with the experiment in which the domain shift is changing colors of in the images (which applies linear transforms to background and foreground pixels), it is hard for me to believe that the same procedure is not tailored for this specific type of domain shift (although this is still an interesting type of domain shift that is important in sim-to-real applications in robotics).

In fact, based on the information bottleneck principle, one could essentially learn that the color is also a feature that also has high MI with label and low MI with inputs, thus the top-left pixel would be indicative of the label. If I use this as the feature for entropy penalty (assuming that there is 1 color per class), this gives a R_{EP} loss of zero. I wonder if this is the reason why the experiments specifically asked for two foreground and two background colors for each class.

Based on there concerns, it might seem more helpful to consider other types of "unknown" domain shifts, such as CIFAR vs. CINIC, in which the method proposed might introduce fewer inductive biases than in the case of being invariant to color.

Minor comments:
	- It would seem like L1 regularization would achieve a similar effect to what is shown in the theory here? It would be interesting to see how sensitivity changes with some existing regularization methods, because they can be easily implemented.
	- Entropy is smaller for lower layers?
	- Minimizing entropy for higher layers at least minimizes an upper bound to the entropy of lower layers (if we consider discrete RVs).
	- Eq 6 is false for continuous RV and differential entropy. One could simply have h_2 = 2 * h_1 to get larger entropy. Eq 5 is true from data processing but the conditional entropy H(h_l | x) is not fixed.
	- It seems that from the discussion about entropy / MI of different layers does not justify the EP approach for the first layer. Even if the final layer retains all the information, you can still technically apply EP to remove redundant information? You can use leaky relu activation to make sure information is not lost in activation functions.
	- Typo for definition of \mu_k
	- The first layer is a convolution layer, so maybe it could be helpful to visualize the learned features with or without EP?


**Experience Assessment:**

I have read many papers in this area.

**Review Assessment: Checking Correctness Of Derivations And Theory:**

I carefully checked the derivations and theory.

**Review Assessment: Checking Correctness Of Experiments:**

I carefully checked the experiments.

**Review Assessment: Thoroughness In Paper Reading:**

I read the paper at least twice and used my best judgement in assessing the paper.

---

### Official Review · AnonReviewer1 · 2019-10-23
**Official Blind Review #1**

**Rating:** 3

**Review:**

This paper proposed Entropy Penalty (EP) training, based on Information Bottleneck (IB) to make the trained model generalize beyond the IID assumption that is satisfied by usual training and testing datasets.

First, a loss function derived from IB is provided Prop. 1. Then they argue that minimizing this loss over lower layers is better Eq. (5) and (6). They further assume the hidden layer is Gaussian, and use squared l2 loss as entropy penalty.

They theoretically analyze a different loss function Eq. (7) under two simple cases, where optimal solutions have closed forms, and find that the optimal solutions contain smaller weights for non-discriminative features (p_i there).

Experiments on coloured versions of MNIST are conducted to show that the proposed Entropy Penalty achieves better results than several baselines.

1. There is no definition of a robust or non-robust feature, but just a vague description by a simple example (camel appearance). I suggest before presenting the method, providing examples like p_i the synthetic cases, to give a sense of what kind of features are (non-)robust.

2. The Gaussian assumption of the hidden layer is quite restricted as mentioned. And why only apply EP on the first layer rather than all the hidden layers?

3. The objective in analysis Eq. (8) looks different from Eq.(7). Why not using the original objective?

4. The synthetic examples are not convincing enough. For example, what are the solutions of other methods, and how can we see the EP solution is better than others for these examples. More comparisons are needed to give a sense of the advantage of EP.

5. Why is randomly assigning colours considered as a distribution shift? It is not clear to connect what the synthetic examples try to deliver and the coloured version of MNIST here.

This paper shows that EP based on IB, together with the Gaussian assumption of the hidden layer can learn robust features as shown in synthetic case analyses and coloured MNIST experiments, comparing with several baselines. However, there are several gaps,

1) what exactly is a non-robust feature, in synthetic examples, these are p_i and k, however, in experiments, it seems to colour. There is no definition and thus it fails to show what exact kinds of features the proposed EP actually can capture (p_i, k, colour, or something more general). In this sense, I have the impression that the presentation is not very clear (EP can learn something, but what it is?).

2) The claim is that EP learns robust features for deep learning methods, but both the analyses and experiments are not enough to show that. First, several restricted assumptions are made as mentioned, learn the first layer only, Gaussian assumption, and the examples are too simple, and lack of calculation for other methods and comparisons. Second, just coloured version of MNIST is not convincing to show that EP captures many/most robust features, even though the proposed EP significantly outperforms the other baselines here because of there probably (must) have many different kinds of other features. More experiments are needed to support the efficacity of EP.

**Experience Assessment:**

I have read many papers in this area.

**Review Assessment: Checking Correctness Of Derivations And Theory:**

I assessed the sensibility of the derivations and theory.

**Review Assessment: Checking Correctness Of Experiments:**

I assessed the sensibility of the experiments.

**Review Assessment: Thoroughness In Paper Reading:**

I read the paper at least twice and used my best judgement in assessing the paper.

---

### Decision · Program_Chairs · 2019-12-19

**Decision:**

Reject

**Comment:**

The paper proposes an entropy penalty related to information bottleneck to deep neural network regression problems. The reviewers had a number of questions and concerns about the paper, which the authors did not address. In light of this, the reviewers agree that the paper is not yet ready for publication. Please carefully read and address the reviewer's concerns in future iterations of this paper.